# Spatial variability of and effect of light on the cœlenteron pH of a reef coral
Lucas Crovetto [1,2], Alexander A. Venn [1], Duygu Sevilgen[1], Sylvie Tambutté [1] ✉ & Eric Tambutté [1]

Coral reefs, the largest bioconstruction on Earth, are formed by calcium carbonate skeletons of corals. Coral skeleton formation commonly referred to as calcification occurs in a specific compartment, the extracellular calcifying medium (ECM), located between the aboral ectoderm and the skeleton. Calcification models often assume a direct link between the surrounding seawater and the ECM. However, the ECM is separated from the seawater by several tissue layers and the cœlenteron, which contains the cœlenteric fluid found in both polyps and cœnosarc (tissue connecting the polyps). Symbiotic dinoflagellate-containing cells line the cœlenteron and their photosynthetic activity contributes to changes in the chemistry of the cœlenteric fluid, particularly with respect to pH. The aim of our study is to compare cœlenteron pH between the cœnosarc and polyps and to compare areas of high or low dinoflagellate density based on tissue coloration. To achieve this, we use liquid ion exchange (LIX) pH microsensors to profile pH in the cœlenteron of polyps and the cœnosarc in different regions of the coral colony in light and darkness. We interpret our results in terms of what light and dark exposure means for proton gradients between the ECM and the coelenteron, and how this could affect calcification.

Coral reefs occupy less than 1.2% of the world's continental shelf area[1], but are of great ecological value. They represent the largest bioconstruction on Earth and host 30% of known marine species[2]. The reefs build up over time, creating a variety of ecological niches necessary for the colonisation and survival of many other marine species. Reef-building scleractinian corals, also called the engineers of the reefs, are calcifying organisms that secrete a calcium carbonate ($CaCO_3$) skeleton through a process known as biomineralization or, more commonly, calcification. Coral skeletons are composite structures containing an organic fraction and a mineral fraction of $CaCO_3$ in the form of aragonite. The precipitation of $CaCO_3$ requires a specific chemical environment at the site of calcification, in which pH is a very important parameter[3]. In the context of current global change, including ocean acidification, many studies have investigated the effects of reduced seawater pH ($pH_{SW}$) on coral physiological processes[4–10].

Detailed descriptions of coral anatomy and histology are reviewed in refs. [3,11]. Briefly, reef-building corals are mainly colonial organisms composed of numerous polyps that are linked together by a tissue called the cœnosarc. The polyp is the anatomical unit of a coral and consists of a central mouth surrounded by a ring of tentacles. Both polyps and cœnosarc contain an internal fluid-filled cavity, the cœlenteron, which in the case of the polyp opens to the external seawater via the stomodeum through the

mouth. The cœlenteron includes the gastrovascular cavity (the interior space of a coral polyp) and the gastrovascular canals (which connect the gastrovascular cavities of polyps). It separates the oral and aboral tissues, which are both composed of an epithelium and an endothelium or gastroderm separated by a layer of extracellular matrix called mesoglea. The oral epithelium faces the surrounding seawater and the oral and aboral endothelia face the cœlenteron. Symbiotic photosynthetic dinoflagellates (family *Symbiodinaceae*[12]) are mostly found in the oral gastroderm and reside within specific cells. The aboral epithelium, also known as the aboral ectoderm or calicoderm, houses the calcifying cells and is located next to the skeleton, playing a key role in its formation. The polyps overlie the calices and the cœnosarc overlies the cœnosteum.

The calcification process takes place in a semi-enclosed compartment located between the calicoderm and the skeleton, namely the extracellular calcifying medium (ECM). Numerous studies have investigated the chemical composition of the ECM, including measurements of pH, calcium, and carbonate concentrations, as these are important parameters controlling the saturation state in the ECM and thus driving calcification[3]. Of these parameters, pH has been the most studied using a variety of approaches. Whether by indirect methods (geochemical proxies[6,13]) or direct methods (pH-sensitive fluorescent dyes[14,15] or pH microsensors[5,16,17]), studies all show

[1]Marine Biology Department, Centre Scientifique de Monaco, 98000, Monaco. [2]Sorbonne Université – ED 515 Complexité du Vivant, 75005 Paris, France. ✉e-mail: stambutte@centrescientifique.mc

 1

that the pH of the ECM ($pH_{ECM}$) is more elevated than pH of the seawater ($pH_{sw}$).

Studies dealing with the calcification process of corals often assume a direct link between the external seawater and the ECM[7,16,18–20]. Recent studies show that in the ECM, calcification involves particle attachment of amorphous calcium carbonate and ion by ion growth[21,22]. However, as described above, the ECM is separated from the external seawater by several compartments, including tissue layers and the cœlenteron. Recently, it has been shown that pH in the aboral mesoglea, which is at the basal side of calcifying cells, has a different pH than seawater[23]. To understand pH gradients across coral compartments (both cellular and extracellular), it is necessary to determine pH values in all of them, including the cœlenteron. The cœlenteron plays a crucial role by serving multiple functions such as digestion, nutrient distribution, waste removal and structural integrity[24]. The cœlenteron could play an important role in mediating the transport of molecules/ions between the external environment, the mesoglea, and the compartment where calcification occurs (i.e. the ECM).

As described earlier, symbiotic dinoflagellate-containing cells line the cœlenteron and their photosynthetic activity contributes to changes in the chemistry of the cœlenteron, particularly with respect to pH. Previous studies using pH microsensors in the cœlenteron focused only on polyps but pH in the cœnosarc has not yet been investigated. Amongst these studies, research using pH microsensors on scleractinian corals have described the variation of pH in the cœlenteron ($pH_{cœl}$) on a daily cycle[4,25–27]. A pH increase is observed in the light due to the photosynthetic activity of dinoflagellates, while a pH decrease is observed in the dark due to the respiration of coral host and symbionts. It is therefore necessary to account for these differences when considering integrated models of physico-chemical gradients between different tissue layers of a coral[28]. Moreover,

although the polyps are connected by the cœnosarc, there are no data in the literature showing whether the composition of the cœlenteric fluid is the same in polyps and cœnosarc.

The pH in the cœlenteron and/or ECM has already been characterised with microsensors in *Montastraea cavernosa, Duncanopsammia axifuga*[4], *Galaxea fascicularis*[25], *Orbicella faveolata, Turbinaria reniformis, Acropora millepora*[29] and with microsensors and pH-sensitive dyes in *Stylophora pistillata*[7,10,14,15,17,30,31], *Pocillopora damicornis*[15] and *Acropora sp*[15,32]. Although data on $pH_{cœl}$ are available for several species[4,25–27,29,33], they were obtained only for polyps. No previous study has addressed the comparison of $pH_{cœl}$ between polyps and cœnosarc or the influence of light intensity or zooxanthellae density within a single coral species. In the present study, we chose to work with *Stylophora pistillata* since it is the coral species in which ECM chemistry has been most extensively studied using geochemical proxies, microsensors, or pH-sensitive dyes[6,10,13,14,17,30,31].

The aim of our study was to determine if the anatomical region (polyp/cœnosarc) and light/dark conditions affect $pH_{cœl}$ and could potentially affect $pH_{ECM}$ in a single coral species. We worked with microcolonies of *Stylophora pistillata* growing on glass slides[34,35]. We used the pH microsensor technique used in ref. [17] for measurements in polyps and cœnosarc with different levels of dinoflagellate density based on tissue colouration (Fig. 1). We first performed depth profiles in the polyps and in the cœnosarc in tissue with a high dinoflagellate density, under light conditions to determine the variation of $pH_{cœl}$. We then measured $pH_{cœl}$ of polyps and cœnosarc at eight light intensities, from darkness to strong illumination, which allowed us to derive a $pH_{cœl}$-irradiance curve and evaluate the role of photosynthesis in influencing pH in the cœlenteron. Finally, we measured $pH_{cœl}$ in the cœnosarc under light and dark conditions in two regions of interest characterised by visually different densities of dinoflagellates

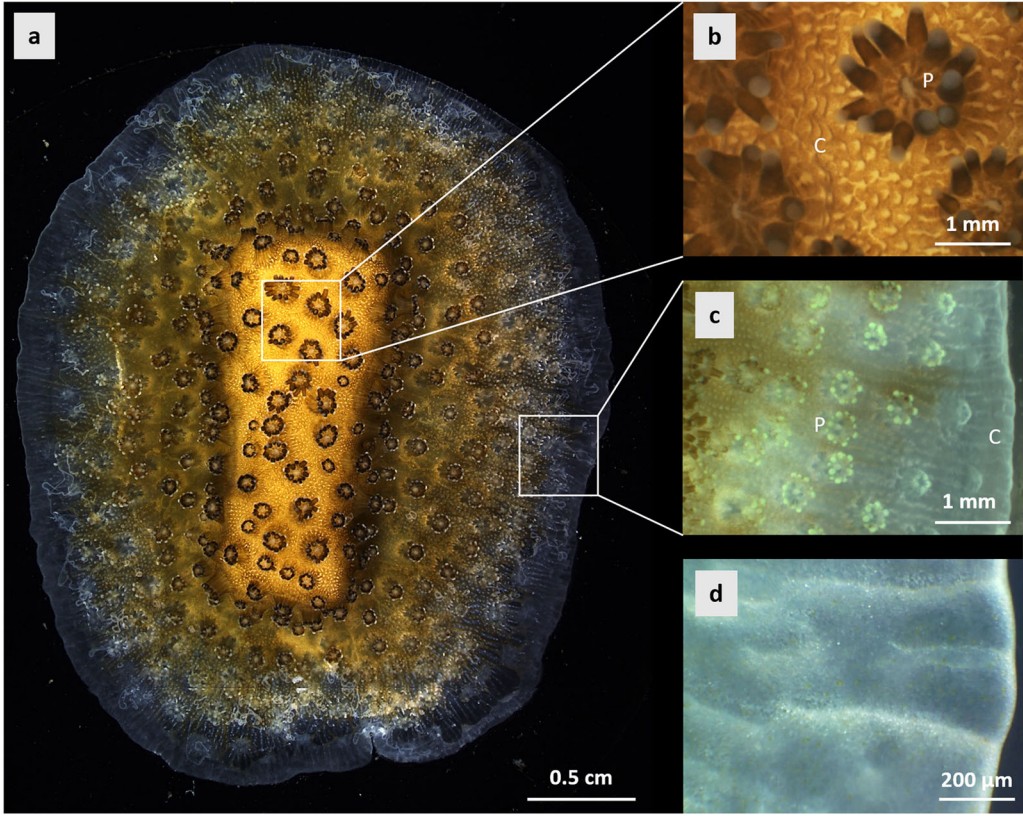

**Fig. 1 | Macroscope images of a microcolony of *Stylophora pistillata* taken from above, looking down on the sample. a** Whole image of a microcolony grown on a glass coverslip. **b** Zoom of the white square in **a** indicating the centre of the microcolony with tissue characterised by a high dinoflagellate density (HDD) (brown tissue). Microsensor measurements of pH in the cœlenteron of HDD tissue were made in such area. **c** Zoom of the white square in **a** indicating the growing edge (GE) of the microcolony with tissue characterised by a low dinoflagellate density (LDD) (transparent tissue). **d** Zoom of the cœnosarc at the growing edge. Microsensor measurements of pH in the cœlenteron of LDD tissues were made in such area. C cœnosarc, P polyp.

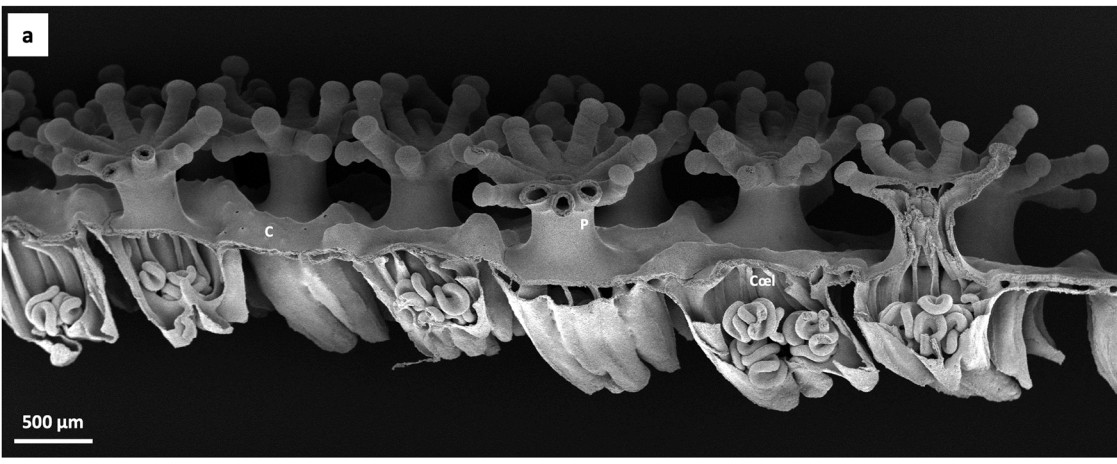

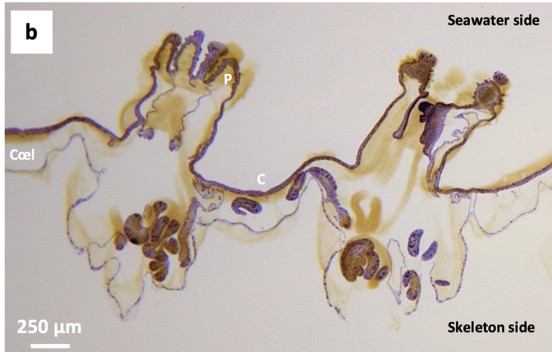

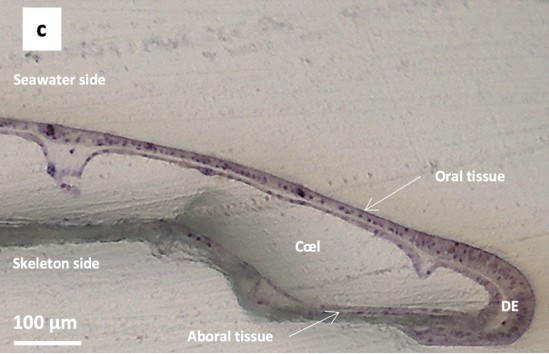

**Fig. 2 | General organisation of the tissue of *Stylophora pistillata* grown on glass slides. a** Decalcified microcolony prepared for scanning electron microscopy observations showing polyps and cœnosarc. **b** Section of a microcolony mounted on a glass slide stained with toluidine blue in borax showing polyps and cœnosarc in HDD tissue. **c** Section of the cœnosarc from LDD tissue, colouration with toluidine blue in borax. C cœnosarc, Cœl cœlenteron, P polyp, DE distal edge.

residing within the coral tissue: a high dinoflagellate density (HDD) tissue at the centre of the microcolonies (Fig. 2a, b) growing on glass slides (brown coloured tissue) and a low dinoflagellate density (LDD) tissue at the edge of microcolonies (transparent tissue), a zone called the growing edge[36] (Fig. 2c).

## Results

### pH depth profiles in tissues with a high dinoflagellate density (HDD) under light

Representative depth profiles for $pH_{cœl}$ of a polyp and cœnosarc in HDD tissue under light are shown in Fig. 3a, b, respectively. Depth profiles of pH in the light (irradiance of 200 µmol photons.m$^{-2}$.s$^{-1}$) were carried out with the microsensor tip progressively inserted through the mouth of the polyp or the tissue of the cœnosarc and then through the cœlenteron until the maximum depth was reached. For the polyp, profiles were stopped when the polyp started to retract due to further advancement of the microsensor. In the cœnosarc, profiles were stopped when the microsensor tip started to lightly bend. All pH data were collected from the surface to the maximum depth by advancing the microsensor downward. During experiments, samples were placed in a temperature-controlled seawater bath (1 L) to maintain a temperature of 25 °C, with a seawater pH of 8.08 ± 0.04 (mean ± SD) (National Bureau of Standards (NBS) scale).

The depth profile of pH in the polyp showed an increase from the mouth (depth 0 µm) with a pH of 8.48, throughout the stomodeum to the entry into the cœlenteron (depth 400 µm) with a pH of 8.69, displaying a Δ pH of 0.21 units relative to the pH at the mouth. From the upper cœlenteron (depth 400 µm) to the bottom of the cœlenteron (depth 1400 µm), the pH stabilises and showed a variation of only 0.07 pH units between the minimum and the maximum value (pH 8.64 and 8.71, respectively).

The depth profile of pH in the cœnosarc showed a similar pattern with an increase from the tissue surface (depth 0 µm) with a pH of 8.37, through the different cell layers of the oral tissue to the entry into the cœlenteron (depth 100 µm) with a pH of 8.73, displaying a Δ pH of 0.36 units compared to the pH at the surface of the cœnosarc. From the upper cœlenteron (depth 100 µm) to the bottom of the cœlenteron (depth 300 µm), the pH stabilised and showed a variation of only 0.06 pH units between the minimum and the maximum value (pH 8.67 and 8.73, respectively).

Overall, these results show that the only difference between the polyp and the cœnosarc lied in the depth at which the cœlenteron was reached. Indeed, the pH exhibited a similar pattern in both the polyp and cœnosarc, with an increase in the first micrometres (i.e. the stomodeum for the polyp and the oral tissue for the cœnosarc) and stabilisation in the cœlenteron itself.

Since pH values in both polyp and cœnosarc remained stable from the top (400 µm for the polyp and 100 µm for the cœnosarc) to the bottom (1400 µm for the polyp and 300 µm for the cœnosarc) of the cœlenteron, we used the mean value of the profile (=$pH_{cœl}$, mean ± SD) to make a comparison between $pH_{cœl}$ in polyp and cœnosarc in the light. Thus, a mean $pH_{cœl}$ value for both the polyp and the cœnosarc was determined for each sample (at least three repeated measurements were made in both polyp and cœnosarc), representing the average of the repeated measurements. Figure 3c shows boxplots with the mean $pH_{cœl}$ of the five different samples of both the polyp and cœnosarc. We found that the mean $pH_{cœl}$ was 8.67 ± 0.27 ($n = 5$) in the polyp, and 8.66 ± 0.18 ($n = 5$) in the cœnosarc, representing a pH variation of 0.59 and 0.58 units, respectively, compared to the $pH_{SW}$. No statistically significant difference in $pH_{cœl}$ was found between these two anatomical regions, but $pH_{cœl}$ was significantly elevated above $pH_{SW}$ in both regions (Wilcoxon test: W = 0, $P < 0.05$).

### Effect of light intensity on pH in the cœlenteron: $pH_{cœl}$-irradiance curve

With $pH_{cœl}$ in the light being stable throughout most of the profile in the polyp and the cœnosarc of HDD tissue (Fig. 3), it allows us to compare these

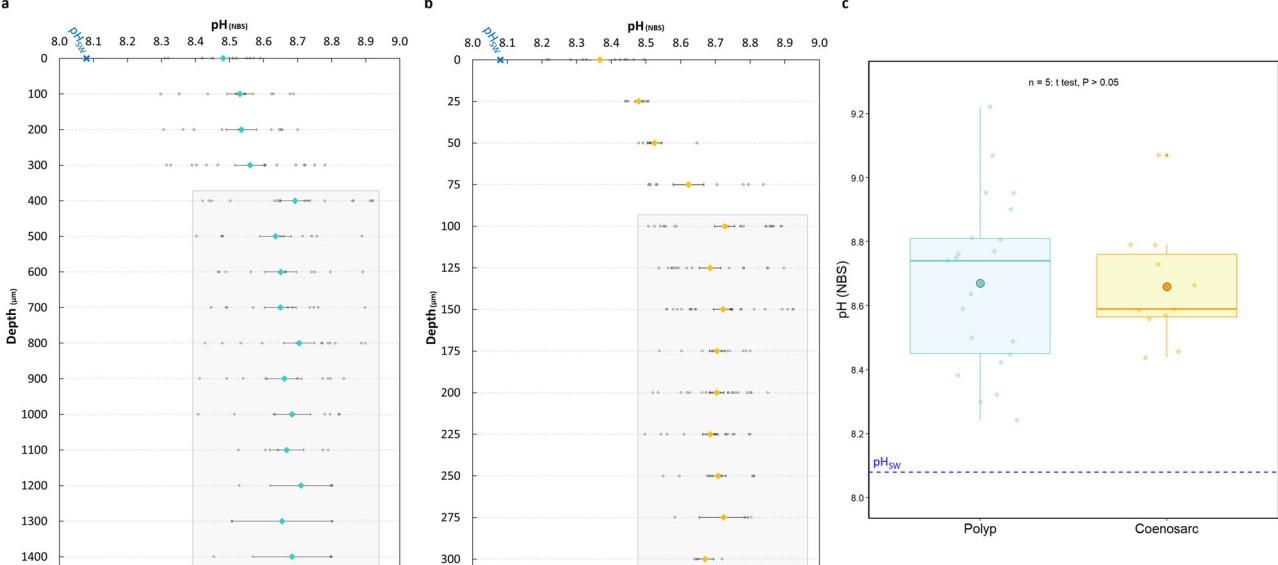

**Fig. 3 | pH measurement in the cœlenteron with a LIX microsensor in polyp and cœnosarc (tissue with a high dinoflagellate density (HDD)) of a _S. pistillata_ microcolony in the light (200 µmol photons.m$^{-2}$.s$^{-1}$).** For clarity, pH depth profiles of polyp and cœnosarc are separated as the depth scales are different (not the same total depth), and individual data points are shown. **a** Representative pH depth profile recorded on a polyp in HDD tissue, considering the sensor tip inserted through the mouth as reference depth 0; depths represent the interior of the polyp from the mouth to reaching the maximum depth (1400 µm); values are mean ± SE calculated from depth profiles made on five different microcolonies, at least three profiles were made per microcolony. Light grey rectangle represents the depths at which the microsensor is inside the cœlenteron. **b** Representative pH depth profile recorded on the cœnosarc in HDD tissue with the sensor tip inserted into the tissue considered as reference depth 0; depths represent the interior of the cœnosarc until reaching the maximum depth (300 µm); values are mean ± SE calculated from depth profiles made on five different microcolonies, at least three profiles were made per microcolony. Light grey rectangle represents the depths at which the microsensor is inside the cœnosarc. **c** Box and whisker plots show the mean (±SD); the first, second (median) and third quartile; and respective whiskers (lowest and highest data point) of cœlenteron pH (pH$_{cœl}$) in the polyp and the cœnosarc; the blue dotted line represents the pH$_{SW}$; paired _t_-test: $t = 0.20365$; df = 3; $P > 0.05$, $n = 5$.

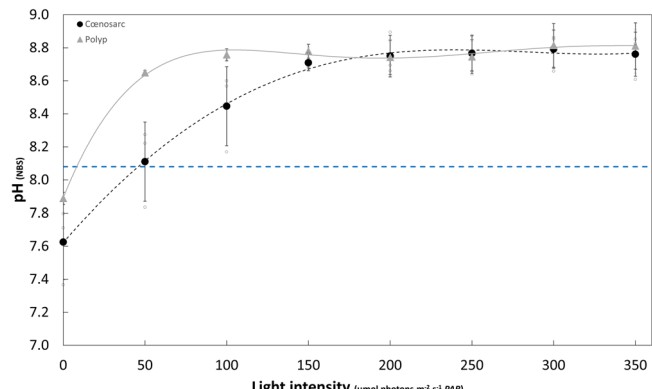

**Fig. 4 | Effect of light intensity on cœlenteron pH (tissue with a high dino-flagellate density (HDD)) of a _S. pistillata_ microcolony sample.** pH$_{cœl}$ values in the polyp (▲) and the cœnosarc (•) of HDD tissue were collected at eight different light intensities during time series lasting between 40 and 60 min, pH values were taken once the signal was stabilised ($n = 3$); data were mean ± SD and individual data points are shown; a polynomial regression curve was fitted to the data for both the polyp and the cœnosarc; the cyan dotted line represents the pH$_{SW}$. Spearman correlation test: Polyp, S = 16.598; $P < 0.05$; rho = 0.80. Spearman correlation test: Cœnosarc, S = 6; $P < 0.05$; rho = 0.93.

two regions at different light intensities. Measurements were carried out at eight light intensities during a time series that lasted between 40 and 60 min (pH$_{SW}$ = 8.08 ± 0.04). At each light intensity except darkness for polyps (see Methods below), we allowed the pH$_{cœl}$ to stabilise for at least 20 min before recording the values. Measurements were made in the polyp and cœnosarc of three samples. Figure 4 shows the pH$_{cœl}$-irradiance curve obtained in the cœlenteron of polyps and cœnosarc in HDD tissue, where pH$_{cœl}$ corresponds to the mean ± SD. pH$_{cœl}$ increased from darkness (pH$_{cœl}$ polyp =

7.89 ± 0.04; pH$_{cœl}$ cœnosarc = 7.63 ± 0.23) until 100 µmol photons.m$^{-2}$.s$^{-1}$ in the polyp (pH$_{cœl}$ = 8.76 ± 0.04) and 150 µmol photons.m$^{-2}$.s$^{-1}$ in the cœnosarc (pH$_{cœl}$ = 8.71 ± 0.05) and then reached a plateau. At lower light intensities (<100 µmol photons.m$^{-2}$.s$^{-1}$) pH$_{cœl}$, even though not statistically significantly different, is slightly higher in the polyps than in the cœnosarc which could be due to a higher density of dinoflagellates per coral biomass. Even though pH$_{cœl}$ plateaued at different light intensities, the values in both the polyp and cœnosarc were stable at higher irradiance. A Spearman correlation test was performed on the data and showed a strong positive relationship between light intensity and pH of the cœlenteron, in both the polyp and cœnosarc (Spearman correlation test: Polyp, S = 16.598, $P < 0.05$, rho = 0.80; Cœnosarc, S = 6, $P < 0.05$, rho = 0.93).

Working at different light intensities also allowed us to determine the optimum irradiance (200 µmol photons.m$^{-2}$.s$^{-1}$, see methods for details) at which to carry out measurements with respect to polyp behaviour and the visibility of the microsensor tip.

## Mapping of pH$_{cœl}$ in cœnosarc of tissue with different dino-flagellate density

Comparisons of cœlenteron pH in tissue with high and low dinoflagellate densities were made exclusively in the cœnosarc, because polyps were not fully formed in low dinoflagellate density regions at the growing edge (Fig. 1b, c).

We first performed a depth profile in the cœnosarc of a low dino-flagellate density (LDD) tissue to determine pH variation. Results show (Supplementary Fig. 1) a decrease in pH from the tissue surface (depth 0 µm, pH = 7.91), to the bottom of the cœnosarc (depth 80 µm, pH = 7.81), displaying a Δ pH of 0.27 units to the external seawater (pH$_{SW}$ = 8.08 ± 0.04). From a depth of 50 µm, pH values stabilised to the bottom of the cœnosarc. Therefore, when measuring pH$_{cœl}$ in LDD tissue, the microsensor was carefully inserted through the tissue and positioned at a depth of 50–70 µm.

pH$_{cœl}$ was measured under light and dark conditions, in seven samples for HDD tissue and five samples for LDD tissue. Each sample was firstly

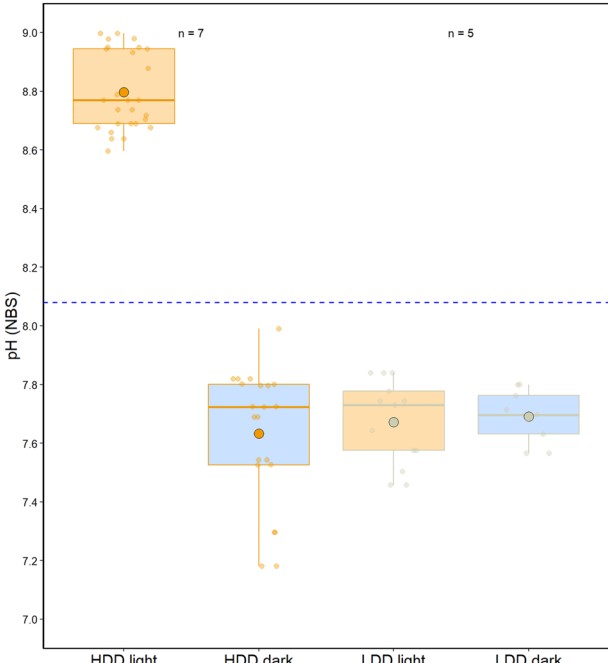

**Fig. 5 | Cœlenteron pH measured in the cœnosarc with high (HDD) or low (LDD) dinoflagellate density of a _S. pistillata_ microcolony in the light and in the dark.** Box and whisker plots show the mean (±SD); the first, second (median) and third quartile; and respective whiskers (lowest and highest data point) of cœlenteron pH (pH$_{cœl}$) obtained on seven samples for HDD tissue and five samples for LDD tissue. Cœnosarc, where measurements were made, is shown in Fig. 1b (HDD) and Fig. 1d (LDD). The blue dotted line represents the pH$_{SW}$. Two-way ANOVA: Area (e.g. HDD or LDD) $F_{1,66}$ = 202.04, $P < 0.05$; Light intensity $F_{1,66}$ = 393.84, $P < 0.05$; Interaction $F_{1,66}$ = 186.02, $P < 0.05$. Letters in superscript indicate subsets determined by Tukey's post hoc analysis.

---

exposed to an irradiance of 200 μmol photons.m$^{-2}$.s$^{-1}$ (light) and then to an irradiance of 0 μmol photons.m$^{-2}$.s$^{-1}$ (dark). For each sample, at least three repeated measurements were performed. For each light condition, pH$_{cœl}$ was allowed to stabilise for at least 20 min before values were taken. Mean pH values (mean ± SD) are shown in Fig. 5 and pH$_{cœl}$ data were compared with a two-way analysis of variance and a Tukey post hoc analysis.

First, regarding HDD tissue, pH$_{cœl}$ in the light was significantly higher than pH$_{cœl}$ in the dark. pH$_{cœl}$ reached a value of 8.80 ± 0.13 under light and 7.63 ± 0.23 under dark, respectively displaying a Δ pH of 0.72 higher than seawater and 0.45 units lower than seawater (pH$_{SW}$ = 8.08 ± 0.04).

Second, regarding LDD tissue, no significant difference was found between light and dark measurements and pH remained stable under all light conditions. pH$_{cœl}$ reached a value of 7.67 ± 0.14 and 7.69 ± 0.09 in the light and dark, respectively, displaying a Δ pH of 0.41 and 0.39 lower than the external seawater (pH$_{SW}$ = 8.08 ± 0.04).

The statistical analysis performed on the data showed a significant effect of the region measured (i.e. the density of dinoflagellates within the tissue) and the light condition alone, and an interaction effect of both parameters on pH$_{cœl}$ (two-way ANOVA: area (e.g. tissue type) $F_{1,66}$ = 202.04, $P < 0.05$; light intensity $F_{1,66}$ = 393.84, $P < 0.05$; interaction $F_{1,66}$ = 186.02, $P < 0.05$). The post hoc analysis revealed two groups: (1) cœnosarc of high dinoflagellate density tissue under light, and (2) cœnosarc of high dinoflagellate density tissue under dark and low dinoflagellate density tissue under both light and dark conditions.

## Discussion

As the largest internal extracellular compartment in corals, the chemistry of cœlenteron is anticipated to influence pH gradients with the ECM where the coral skeleton forms. In the current study, we focused on the pH of the cœlenteric fluid because pH is a major parameter affecting coral calcification, together with other parameters, including calcium and dissolved inorganic carbon concentrations. In colonial corals, the cœlenteric fluid is found not only in the polyps but also in the connecting tissue, the cœnosarc (Fig. 2). In the present study, we measured pH$_{cœl}$ in different anatomical regions of the coral: polyps and coenosarc and investigated the influence of dinoflagellate density (observed by tissue colouration) in both light and dark conditions in the cœnosarc.

pH depth profiles performed in the polyps of _S. pistillata_ microcolonies are consistent with the previous study of ref. [17], which used a similar experimental set-up (specifically similar seawater, coral species, feeding and light conditions). The increase in pH with depth and stabilisation of values in the cœlenteron (or gastrovascular cavity for some references listed) with little pH variation (Fig. 3) was also observed in other coral species such as _Acropora sp._, _Favia sp._, _Orbicella aveolate_ and _Turbinaria reniformis_[26,29,33]. Although pH rises similarly in the first micrometres after insertion of the microsensor tip through the mouth and then stabilises in the cœlenteron, the depths at which this compartment is reached is species-specific. In _Turbinaria reniformis_ and _Acropora millepora_, the increase in pH occurs from a depth of 400–500 μm[29,33] and is similar to _S. pistillata_ (400 μm; present study), while in _Favia sp._, the pH increases rapidly after entering the polyp mouth and reaches its maximum value at a depth of about 300 μm[26]. These differences in pH depth profiles are probably linked to the tissue/skeletal relationship in these various species, with some belonging to the "Complex" and other to the "Robust" clade. The latter, in which _S. pistillata_ belongs, presents heavily calcified skeletons whereas the "Complex" corals (e.g. _Acropora sp._) tend to be less heavily calcified[37]. This could have an influence on the fluid chemistry of the gastrovascular cavity with polyps being more or less isolated from each other. In this study, we only focus on _S. pistillata_ from the "Robust" clade, but it would be interesting to make a comparative study on coral species belonging to the different clades in future work.

The pH depth profiles performed in the cœnosarc of a microcolony of _S. pistillata_ were similar to those of the polyps, only the depth at which the cœlenteron was reached differs (100 and 400 μm for cœnosarc and polyp, respectively) (Fig. 3a, b). This result is not surprising and can be explained by looking at the anatomy of _S. pistillata_. Indeed, the cœlenteron of the polyp is much deeper than the cœlenteron of the cœnosarc relative to the surface of the coral.

Previous studies on the cœlenteron using pH microsensors found that, on a daily cycle, pH$_{cœl}$ increases in the light due to photosynthesis, while pH$_{cœl}$ decreases in the dark due to respiration[4,25–27]. In addition, there is a positive relationship between coral photosynthesis and calcification under light conditions, a process known as light-enhanced calcification (LEC). There are many hypotheses to explain LEC, and one of them involves the increase of pH in the cœlenteric fluid, which favours the removal of protons from the ECM[38–40]. Many studies have investigated the photosynthesis-irradiance (PI) relationship in corals and have characterised how photosynthetic rates increase with increasing light intensity until a plateau is reached[40–42]. However, although pH$_{cœl}$ is often assumed to be light dependent[4,25,26], the pH$_{cœl}$-irradiance relationship has never been determined, and therefore the full range of pH$_{cœl}$ has remained uncharacterised. Therefore, we used multiple light intensities to measure pH$_{cœl}$ in both polyps and cœnosarc in HDD tissue. The resulting pH$_{cœl}$-irradiance curve showed a strong positive relationship between pH and light intensity, reflecting the photosynthetic activity of the symbionts and its effects on pH cœlenteron and provides us with the full range of light-driven pH$_{cœl}$ changes (Fig. 4).

As mentioned above, symbiotic dinoflagellates are not evenly distributed throughout the coral tissue in _Stylophora pistillata_. Unlike the white-transparent tissue observed in bleached corals[43], such tissues are also observed in non-stressful conditions in active growing zones such as at the tip of coral branches[44,45]. This is also the case at the growing edge prepared with the lateral preparative assay[14,46], and this can be clearly seen in Fig. 1a, c, d. Knowing that pH$_{cœl}$ is directly influenced by the photosynthetic activity

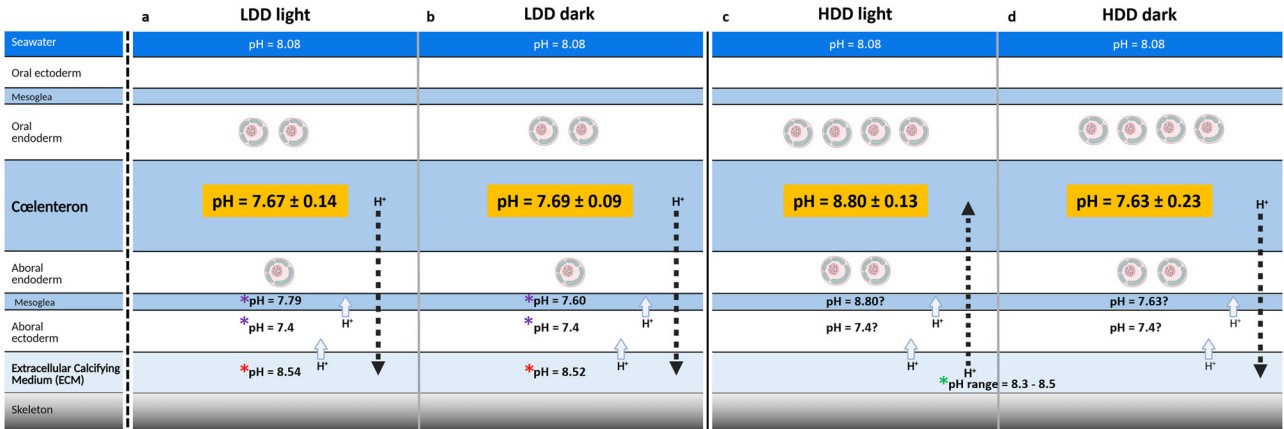

**Fig. 6 | Diagram depicting a model of the influence of light and dark on proton gradients across the tissue layers of a microcolony of *S. pistillata*. a, b** represent low dinoflagellate density (LDD) tissue in light and dark conditions, respectively. **c, d** represent high dinoflagellate density (HDD) tissue in light and dark conditions, respectively. pH values in orange correspond to cœlenteron pH measured in the present study. ＊ $pH_{ECM}$ (total scale) from skeletal boron isotope analysis[6,13,51,52].

＊ pH of the mesoglea and aboral ectoderm measured with pH-sensitive fluorescent dye from ref. [23]. ✳ $pH_{ECM}$ measured with microsensors from ref. [17]. Symbiotic dinoflagellates are represented in gastrodermis for both LDD and HDD tissues. Dashed arrows represent the paracellular pathway along the concentration gradient between the cœlenteron and the ECM. Light blue arrows represent the removal of $H^+$ from the ECM via active transcellular mechanisms.

of dinoflagellates, we aimed to compare $pH_{cœl}$ in high and low dinofloagellate density tissue under light and dark conditions. The $pH_{cœl}$ in HDD tissue showed values above that of the external seawater under light conditions where photosynthesis occurs and below that of the surrounding seawater under dark conditions where respiration of both host and symbionts occurs (Fig. 5). The results of this study are consistent with previous studies that have shown shifts in $pH_{cœl}$ between light and dark conditions[4,25,26,47]. However, previous studies did not characterise full $pH_{cœl}$-irradiance relationships, so it is not known if previous reports of light-driven $pH_{cœl}$ changes represent the full range of $pH_{cœl}$. When measured in LDD tissue (Fig. 5), $pH_{cœl}$ remains stable in both light ($pH_{cœl} = 7.67 \pm 0.14$ at 200 μmol photons.m$^{-2}$.s$^{-1}$) and dark conditions ($pH_{cœl} = 7.69 \pm 0.09$ at 0 μmol photons.m$^{-2}$.s$^{-1}$), indicating that pH is kept below $pH_{SW}$ ($pH_{SW} = 8.08 \pm 0.04$) when net respiration of both host and symbionts occurs. Even though low numbers of dinoflagellates are present at the growing edge, their combined photosynthetic activity is not sufficient to increase $pH_{cœl}$ in the light. It is noteworthy that the values of $pH_{cœl}$ in LDD tissue are similar to those measured under dark conditions in HDD tissue, suggesting that $pH_{cœl}$ in this latter tissue does not decrease below a threshold value of ~7.6. Such a result could be explained by similar production of $CO_2$ by respiration in the two zones and presumably similar rates at which $CO_2$ diffuses to the surrounding seawater. Also, the difference in pH in the light between $pH_{cœl}$ in the coenosarc at the centre of the colony (HDD region) and the growing edge (LDD region) suggests, that cœlenteron fluid circulation in the colony is not sufficient to lead to light-driven increases in $pH_{cœl}$ at the growing edge.

Our results show that $pH_{cœl}$ in LDD tissue (at the growing edge) is lower than $pH_{ECM}$ determined in previous studies[17] in light and darkness in *S. pistillata* (Fig. 6). Moreover, our $pH_{cœl}$ values are similar to the pH values determined previously in the mesoglea, which lies between the cœlenteron and ECM[23]. Together these data suggest that paracellular diffusion of protons from the ECM to the cœlenteron is unlikely to occur as it is against the concentration gradient (Fig. 6). Instead, active transcellular mechanisms must be involved to move protons out of the ECM via membrane transporters of the calicoderm[48–50]. We recognise that Venn et al. [23] recorded a small difference in the pH of mesoglea between light and dark conditions, and we did not measure light/dark differences of $pH_{cœl}$ in LDD tissue in our current study. This inconsistency might be due to a difference in symbiont density at the growing edge in the samples used in the two studies.

In HDD tissue, our dark measurements are similar to the values obtained at the growing edge (LDD) in both light and darkness. Direct measurements of $pH_{ECM}$ have never been achieved with optical verification

in the zone of high symbiont density but estimates of $pH_{ECM}$ by skeletal boron isotope analysis (that do not discriminate between light and dark conditions) in *Stylophora pistillata* and other corals indicate that $pH_{ECM}$ is in the range of pH 8.3 to 8.5 (total scale)[6,13,51,52] (Fig. 6). Contrary to mesoglea pH in LDD tissue (at the growing edge[23]), mesoglea pH has not been measured previously in HDD tissue in either light or dark conditions. Since $pH_{cœl}$ was similar to pH mesoglea in the LDD tissue, we assume that this is the same in the HDD tissue. As such, there is likely to be an unfavourable gradient of protons from the ECM to the cœlenteron in the dark in this zone. Indeed, paracellular diffusion of protons away from the ECM is unlikely to occur in these circumstances, and active transport mechanisms are likely to be required to maintain elevated $pH_{ECM}$ relative to the cœlenteron and seawater.

In HDD tissue in the light, our measurements indicate that $pH_{cœl}$ is higher than previous estimates of $pH_{ECM}$ in this zone (values by skeletal boron isotope analysis as above, Fig. 6). In contrast to dark conditions and both light and darkness in LDD tissue, the pH gradient is favourable for the diffusion of protons from the ECM into the cœlenteron. We propose that paracellular and transcellular mechanisms may operate in parallel to move protons produced by calcification in the ECM across the calicoblastic epithelium to the cœlenteron. In the cœlenteron, protons are then neutralised by reaction with OH⁻, released by photosynthesis[38,53,54].

Our results and their interpretation in the diagram in Fig. 6 agree with one of the previously proposed mechanisms for light-enhanced calcification[38,40]. Under light conditions, elevated $pH_{cœl}$ driven by symbiont photosynthesis in HDD zones would enhance proton flux from the ECM. If this resulted in higher $pH_{ECM}$, then this could increase the $CaCO_3$ saturation state and thus lead to higher rates of precipitation. Similarly, higher $pH_{ECM}$ would presumably also be favourable to higher pH in macropinocytotic vesicles that engulf ECM, thus potentially favoring the formation of intracellular ACC (amorphous calcium carbonate) precursors in the calicoblastic cells[21]. It is worth pointing out that higher rates of precipitation would result in higher proton production and therefore decrease pH. As such, $pH_{ECM}$ values found in HDD tissue may not be markedly higher than LDD tissue like the growing edge, as calcification rates may be higher. Further research is required to explore this issue. Changes in proton gradients may also modify membrane/transepithelial potential, thus influencing ion transport across membranes. However, it has been shown that the light-mediated electrical potential is independent of the photosynthetic activity of the algal symbionts[55], thus ruling out this possibility. It is also important to point out that other mechanisms underlying LEC may also operate, including greater energy supply from photosynthesis for active ion

transport[56] and the supply of organic matrix precursors from the symbionts[57].

The present study showed the importance of $pH_{coel}$ in proton gradients between the different coral's compartments and its implication on the calcification process. However, measurements were performed under controlled conditions and did not look at the impact of seawater acidification on the cœlenteron and its possible impacts on gradients between the surrounding seawater and the coral. A study performed on *M. cavernosa* and *D. axifuga* showed a species-specific response to a decrease in seawater pH but focused only on cœlenteron pH[4]. This study suggests that the photosynthetic activity of symbiotic dinoflagellates can partially mitigate the negative effects of ocean acidification on calcification rates. In *S. pistillata*, previous studies focused on the effects of seawater acidification on the pH of the ECM[7,15,23]. These studies have shown that ocean acidification has a major impact on coral physiology, but the effects depend on the species, light and compartment studied. The ECM is relatively well regulated with respect to pH, but mesoglea is more pH-conforming with respect to the external seawater environment. However, the effects of ocean acidification on cœlenteron pH remain unknown. This is an important area for future research as the cœlenteron could act as a buffering compartment that mitigates the effects of decreasing $pH_{SW}$ and helps maintain a favourable chemical environment for calcification in the ECM.

In summary, the present study characterised the $pH_{coel}$ of *S. pistillata* in both polyps and cœnosarc and in tissue with different dinoflagellate densities under light/dark conditions. The $pH_{coel}$ of HDD tissue exhibits light/dark fluctuations due to the photosynthetic activity of the symbionts. By contrast, the $pH_{coel}$ of LDD tissue measured in the cœnosarc does not exhibit light/dark variations and the pH values measured here are much lower than those of seawater and ECM. When $pH_{coel} < pH_{ECM}$ (Fig. 6a, b, d), paracellular diffusion of protons from the ECM to the cœlenteron is unlikely to occur as it is against the concentration gradient. Active transcellular mechanisms must be involved to move protons out of the ECM via membrane transporters of the aboral ectoderm. When $pH_{coel} > pH_{ECM}$ (Fig. 6c), the concentration gradient could be favourable for diffusion of $H^+$ from the ECM into the cœlenteron.

The inclusion of the cœlenteron in calcification models is imperative, with particular attention to its chemical composition, especially in terms of pH. The importance lies in the efficient removal of protons from the calcification site. However, pH is only one parameter that influences calcification. For a comprehensive understanding of the cœlenteron carbonate chemistry, including factors such as carbonate and calcium concentration, additional experiments are needed. Furthermore, research into the effects of environmental factors, such as seawater acidification, is crucial for a more sophisticated understanding of the calcification process.

## Methods

### *Stylophora pistillata* microcolonies
*S. pistillata* colonies, maintained at the Centre Scientifique de Monaco, were used to produce microcolonies grown on glass slides according to the technique initially described by ref. [34], later referred to as the lateral skeleton preparative assay ref. [35], and since then used in many physiological studies[7,10,14,17]. Briefly, pieces of microcolonies were cut with a razor blade and fixed with resin (Devcon™) on rectangular glass slides. These pieces were then left to grow (Fig. 1a) in long-term coral culture facilities supplied with flowing seawater from the Mediterranean Sea (exchange rate 170%.h$^{-1}$), at a salinity of 38, temperature of 25 °C, under an irradiance of 175 µmol photons.m$^{-2}$.s$^{-1}$ (provided by a BLV HQI Light Bulb Nepturion, 150 W) on a 12 h: 12 h photoperiod. Corals were fed both with frozen rotifers (daily) and live *Artemia salina* nauplii (twice per week). During experiments, samples were placed in a temperature-controlled seawater bath (1 L) to maintain a temperature of 25 °C, with a seawater pH of 8.08 ± 0.04 (mean ± SD) (National Bureau of Standards (NBS) scale), and light intensities ranging from 0 to 350 µmol photons.m$^{-2}$. s$^{-1}$ (provided by a CL 9000 LED lamp, Zeiss© , Germany and measured with a Walz US-SQS/L Submersible Spherical Micro Quantum Sensor, Heinz Walz GmbH©,

Germany). The seawater bath was filled with water from the coral culture aquaria during experiments.

### Microsensor construction and calibration
pH-liquid ion exchange (LIX) microsensors were prepared as described previously[17]. Briefly, glass capillary tubes (borosilicate; 8 cm length; 1.5 mm diameter; Science Product, Germany) were pulled on a DMZ Universal puller (Zeitz Instruments). The micropipettes with a tip diameter of 2–5 µm were silanized and backfilled with electrolyte specific for $H^+$, then frontfilled with the LIX membrane containing the $H^+$ ionophore and let to dry for several hours to allow them to stabilise prior to measurements. Several microsensors were manufactured at once to have a stock in case of malfunction.

Calibration of the pH microsensors was performed in seawater adjusted to pH 7 to 9 by adding HCl and NaOH in 0.5 pH units (NBS scale) as described previously[17]. The pH of seawater was measured using a pH electrode (Mettler Toledo) previously calibrated with three commercially available pH NBS buffers (pH 4, 7, 10; Hannah Instruments Buffer Solutions).

### Experimental set-up
All experiments were performed under a Leica Z16 APO macroscope (Leica Microsystems) connected to a camera system and a computer monitor that allowed live macroscopic observations (Archimed® Microvision, France). Macroscope images of the insertion of a microsensor tip in an *S. pistillata* microcolony are shown in Supplementary Fig. 2. The use of a motorised micromanipulator allows precise movements of the microsensor on the order of micrometres (MUX2, PyroScience GmbH, Germany). The set-up used in this study is the same as that used ref. [17] for depth profiles of pH obtained on polyps of *S. pistillata*. Microsensor signals were recorded every 5 s.

### Light microscopy
Coral samples growing on a coverslip are fixed overnight in 4% glutaraldehyde in artificial seawater buffered to pH 7.8 with 0.1 M sodium cacodylate (according to ref. [34]). The samples are then rinsed in distilled water before being dehydrated through a series of ethanol solutions. The coral was then embedded in EPO-TEK® (Epoxy Technology, France) for sectioning. The sections (1.0 mm) were cut using the Minitom® and a diamond cut-off wheel Minitom® (Struers, France). The section was mounted on glass slides, polished using silicon carbide foils (up to 4000 grades, lubricated with water), and stained with toluidine blue in borax and photographed with a Leica DM750P.

### Scanning electron microscopy
Samples of *S. pistillata* growing on a glass slide were processed as described in ref. [3]. Briefly, samples were fixed overnight at 4 °C with 4% glutaraldehyde in 0.085 M Sorensen phosphate buffer at pH 7.8 with 0.5 M sucrose. Decalcification was achieved by transferring the samples to a mixture of 0.085 M Sorensen phosphate buffer, 0.5 M sucrose containing 2% glutaraldehyde and 0.5 M ethylenediaminetetraacetic acid (EDTA) at pH 7.8 and 4 °C. This solution was renewed until decalcification was completed. Decalcified samples were rinsed in Sorensen buffer, then post-fixed for 1 h at ambient temperature with 1% osmium tetroxide in Sorensen phosphate buffer. Samples were dehydrated by transfer through a graded series of ethanol ending with a concentration of 100%. After dehydration, they were incubated for 15 min in hexamethyldisilazane (HMDS)/ethanol 100% (v/v), then 30 min in HMDS 100% that was subsequently evaporated under a fume hood overnight. Samples were then coated with gold-palladium and observed at 3–5 kV with a JEOL JSM-6010LV.

### pH depth profiles: polyp and cœnosarc with high dinoflagellate density
Depth profiles were performed only under light conditions (irradiance of 200 µmol photons.m$^{-2}$.s$^{-1}$) in polyps and cœnosarc of tissue with a high

dinoflagellate density (Fig. 1b) to determine the variation of cœlenteron pH in *S. pistillata*. For the polyp profiles, the tip of the microsensor was positioned above the mouth, corresponding to what we defined as depth 0 (μm). The microsensor was then inserted through the mouth until the polyp began to bend and retract. We took this depth as the maximum depth. Profile data were collected from the mouth of the polyp in incremental steps of 100 μm downward to the maximum depth (bottom of the polyp). For profiles in the cœnosarc, the tip of the microsensor was positioned at the tissue surface, corresponding to a depth of 0. The microsensor was inserted through the tissue until the tip of the microsensor began to lightly bend. We took this depth as the maximum depth. As with the polyp profiles, data were collected from the tissue surface in incremental steps of 25 μm as we moved downward to the maximum depth (bottom of the cœnosarc). Insertion of the microsensor tip into the polyp and cœnosarc was checked both visually by macroscopy and by a sudden change in the signal. To compare the pH$_{cœl}$ of polyp and cœnosarc under light conditions, we used the mean value of the profile (=pH$_{cœl}$) corresponding to the depths at which pH values remained stable along the depth profile through the cœlenteron.

### Effect of light intensities on pH in the cœlenteron (HDD tissues)

pH microsensor was positioned at a depth corresponding to stable pH$_{cœl}$ in both the polyp and cœnosarc of HDD tissues, and measurements were made at eight different light intensities ranging from strong illumination to darkness: 350, 300, 250, 200, 150, 100, 50 and 0 μmol photons.m$^{-2}$.s$^{-1}$. For each light intensity, values were recorded from an entire time series that lasted between 40 and 60 min. pH$_{cœl}$ for each time series were averaged after the microsensor readings stabilised. In polyps, darkness causes the polyp to retract into the corallite calyx, causing the microsensor to exit it and alter the signal. Therefore, pH$_{cœl}$ in the polyp under dark conditions corresponds to the stabilised microsensor readings prior to this complete retraction of the polyp. In addition to pH$_{cœl}$, we also evaluated the behaviour of the polyps (whether they retracted or not), and the resolution of the macroscopic observations (whether the tip of the microsensor was clearly visible or not).

### Mapping of cœlenteron pH in tissues with high and low dinoflagellate density

We performed measurements under light and dark conditions in the cœnosarc in two regions of interest characterised by visually different densities of dinoflagellates residing within the coral tissue: tissue with a high dinoflagellate density (HDD) versus tissue with a low dinoflagellate density (LDD) (Fig. 1b, d). For the region characterised by a low dinoflagellate density (transparent tissue), depth profiles were carried out to determine the depth of stable pH$_{cœl}$ values as was performed for polyps and cœnosarc in HDD tissue (see Methods above). The microsensor was inserted through the tissue until the maximum depth was reached. Data were collected from the tissue surface to the maximum depth (bottom of the cœnosarc) in incremental steps of 10–20 μm, moving downward. The microcolonies were selected so that the growing edge was wide enough for access with microsensors. pH$_{cœl}$ was measured within the first 400 μm from the edge of the sample, as has been done previously with confocal microscopy[14,17]. In this area, the oral and aboral epithelia (including the calicoderm) are present (Fig. 2), and a new skeleton is in the process of forming[14]. Once the depth of measurement was determined, for each region of interest (cœnosarc with high or low density of dinoflagellates), measurements were made at an irradiance of 200 μmol photons.m$^{-2}$.s$^{-1}$ (light) and 0 μmol photons.m$^{-2}$.s$^{-1}$ (dark). For each sample, pH$_{cœl}$ was recorded first under light and then under dark conditions during a time series of 40 to 60 min. The first 20 min of each time series were discarded to allow stabilisation of the signal. After the dark period, the light was turned back on for at least 10 min before the microsensor was removed and positioned in seawater. As previously described, the pH$_{cœl}$ values for each time series were averaged for each replicate after the microsensor readings were stabilised. pH$_{cœl}$ replicates were obtained in separate samples and averaged (±SD) for final values.

### Statistics and reproducibility

Seven samples of *S. pistillata* grown in long-term coral culture facilities on glass slides at the Centre Scientifique de Monaco were used for this study. For each sample, at least three replicate measurements were performed under all conditions to allow statistical analysis if required. Calibration curves, graphs and statistical analyses were performed using Excel and the software RStudio[58]. Spearman correlation test, *T*-tests, and two-way analyses of variance (ANOVA) were performed on the data. Post hoc analyses were also performed as needed. All statistical analysis performed in the current study are shown in Supplementary Tables 1, 2.

### Reporting summary

Further information on research design is available in the Nature Portfolio Reporting Summary linked to this article.

### Data availability

Numerical source data for graphs and charts can be found within the Supplementary Data file. Additional information and relevant data will be available from the corresponding author upon reasonable request. The datasets presented in this study can be found in the online repository: https://www.pangaea.de/tok/358beef9d2a11b64a8f5e964d6543b1f49ba056b.

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

## Acknowledgements
We would like to thank Dominique Desgré and Xavier Maccario for coral maintenance. This study was conducted as part of the Centre Scientifique de Monaco research programme, supported by the Government of the Principality of Monaco.

## Author contributions
L.C. and E.T. carried out research, contributed to the conception and design of the study and analysed data. L.C., A.V., S.T. and E.T. wrote the manuscript. All authors contributed to the article and approved the submitted version.

## Competing interests
The authors declare no competing interests.
