## [Peer Review File · Communications Biology]

Reviewers' comments:

Reviewer #1 (Remarks to the Author):

Review of: Spatial variability and effect of light on coelenteron pH of a reef coral, by Crovetto et al
In this study, the authors make pH measurements with microelectrodes in various locations within microcolonies of *Stylophora pistillata*, with the goal of differentiating the effects of light and dark on areas of the colony with higher or lower densities of zooxanthellae (not quantified, just a visual more or less brown), and also inside polyps and coenosarc within the central area of the microcolony where the polyps are growing vertically (and thus lots of skeletogenesis occurring) and the thin coenosarc skirt of the colony where there is primarily tissue extension taking place, without polyps and little to no calcification (not measured).

I think the technical aspects of the work are solid and seem to have been carefully executed (and this is painstaking, difficult work), but the write-up and interpretation of the results needs a lot of revision. I made a lot of notes and edits in the pdf, which I will not list in detail here (the authors can go through them and see what needs to be addressed). So here my review will focus on the major interpretation flaws that I suggest the authors concentrate on in their revision.

1) Terminology: Ectoderm and endoderm are developmental terms. Epiderm and endoderm and variants (e.g. calicoderm, gastroderm, epithelia) are the terms the authors need to substitute throughout the paper. I pointed out many but not all incidences of misuse.

2) Coelenteron = gastrovascular cavity (GVC) preferred term, is located only within the body of the polyp. The tubules/channels/extensions of the GVC that connect GVCs of adjacent polyps are not technically part of the coelenteron/GVC.

3) When thinking about their results the authors need to consider that corals are currently grouped into two major clades: Robusta and Complexa. Coral colonies in the Robusta clade (in which group *Stylophora* is placed), have well defined polyps and calices with thicker cell walls (theca) and the polyps are only connected to each other by the thin layer of coenosarc that sheets out at the tops/oral ends of the polyps. A decalcified piece of such coral looks like a sheet of naked polyps protruding downwards from the surface coenosarc that connects them (like a six pack of soda cans suspended by those plastic rings). Most of the skeletal structure in the Robusta is secreted by the calicoderm/calicoblastic epithelium located within the polyps. Thus, the GVC and its fluids are more isolated from polyp to polyp, the fluid chemistry will be more affected by the metabolic activity of that particular polyp, and this chemistry is very important to skeletogenesis in these corals. The thin coenosarc between the polyps produces the coenenchyme, which is much less dense skeletal material than the polyp calices. But the coenosarc has to keep secreting new layers of dissepiments so that the skeleton between the polyps can keep up with skeletal growth by the polyps. Corals in the Complexa clade have poorly defined corallite walls, and extensions from the GVC that connect adjacent polyps occur top to bottom of the polyp. A decalcified piece of a Complexa colony looks more like a kitchen sponge where the polyp mouths slightly protrude from the surface of the spongy layer, and the aboral ends of the polyps are embedded in the spongy material. Thus, there is much less isolation of the GVCs of adjacent polyps and while I have not seen measurements published, I assume there is a lot of fluid interconnection among polyps.

4) Now let's think about the microcolony structure: when a planula settles, it first secretes a thin organic adhesive to attach to the substrate as it morphs into the primary polyp. All of this happens before any calcification starts, and in fact before the calicoderm differentiates. Then over days, mesenteries, a primary base plate and theca, and primary skeletal septa develop. The skirt of soft tissue around the primary polyp expands over time if the polyp has enough nourishment, and eventually more polyp mouths develop on the upper surface of this skirt. Thus, first there is tissue attachment, then tissue remodeling, then calcification/skeletogenesis. Basically, the same happens when the microcolony preps are set up. The nubbin develops a tissue skirt (coenosarc without polyp mouths and no calcification), then as the tissues differentiate you see tiny mouths pop out but without any aboral depth and little to no calcification, and finally the new skirt grows and differentiates, calicoderm forms and calcification begins. Apply this to where the present researchers took pH measurements, the HDD ones were from within polyps or coenosarc from the original nubbin (mature tissue differentiation), while the LDD ones were from the newly forming skirt without much biomass, no polyps or GVC, and most likely little to no calicoderm differentiation (would need histology to confirm).

I make this point about the body plan and development of the species used in this study because if the authors rethink the message they are trying to present, I think their data will make for a more

important and novel story. Just being the first to measure pH in the HDD vs LDD of the coenosarc of this species, whether in the light or the dark, doesn't really help us understand anything new. This was the question I kept asking myself as I read the paper: so what?

Here is how I analyzed their results:

1) As previously demonstrated in a number of studies, GVC pH increases in the light (presumably due to photosynthesis by algal symbionts) and decreases in the dark as CO₂ and other metabolites accumulate in the GVC. Nothing new here but adding the P-I curves is very nice.

2) Measuring the same light-dark pattern in the HDD coenosarc surrounding mature polyps demonstrates that the same tissue chemistry is happening in the areas in between polyps. Not a surprising result, and in fact Justin Ries used microelectrodes to measure pH in *Astrangia poculata* in 2008 and only measured between the polyps. But still nice to know that the effect of the zooxanthellae on internal pH of coral tissues. Further, much more coral animal biomass and more zoox inside the polyp so pH reached slightly higher level and closer to the mouth opening than in coenosarc. See my comment in Figure 3 below.

3) What I found more interesting, and novel was that in newly developing and poorly differentiated coral tissues, such as where they made their LDD measurements, basically there was no effect of light (because few zooxanthellae) but also most likely no calcification. Not much coral biomass per area in this region of the microcolony. But without photosynthesis, tissue/fluid pH was negative all the time. It would have been interesting if they had done a transect from the growing edge back towards the nubbin center and correlated it with the tissues differentiating to acquire zooxanthellae and the development of calicoderm and calcification.

Again: Many issues that the authors need to address as indicated by comments and highlights in the PDF text, which I will not list here.

Regarding data presentation and manuscript preparation: The English needs work. I hope that one of the senior co-authors who has a better command of English can help fix many unclear and clumsy sentences and paragraphs.

The discussion goes on and on, unfocused, and repeats material that should have been in methods or results.

The seawater pH in which measurements were made needs to be specified in many places and in the figures, not just as a detail in the methods.

Fig 2: Font of axis labels is way too small. Initial SW pH not shown. Use the same depth scale for the polyp and coenosarc vertical transects. Remove panel (C) and include in Figure 4.

Fig 3: The coenosarc dark and low light pH were quite a bit lower than the polyp pH. I would note this in the results. Suggests to me that the coenosarc had lower zoox density per coral biomass than in the polyp proper, and maybe only at higher light levels was pH in these inter-polyp areas affected by fluids flowing from the polyp GVC.

Fig 4: Add panel (C) from Fig 2 so that all the similar type of data are together.

Fig 5: Rethink this model and tissue characteristics based on what I explained above. The areas where LDD measurements were made have little to no calcification going on, and I would bet, do not have a differentiated calicoderm layer, thus no calcification, ion fluxes etc.

Reviewer #2 (Remarks to the Author):

The work of Crovetto and co-workers reports the differences in the pH of the coelenteron of the polyps (also known as gastric cavity) and the coenosarc of microcolonies of the coral *Stylophora pistillata* with high and low endosymbiont density. Authors used pH microsensor profiles to locate the coelenteron and performed light curves to identify the proper experimental set up for later experiments. The manuscript is well written and results are discussed in light of our current knowledge of the pH dynamics of the different layers of the coral tissue coupled with a very informative summary diagram at the end. However, before publication the authors should consider the comments outlined below.

Introduction:

- The "polyp coelenteron" is also called, in recent literature, the polyp "gastric cavity". It would be recommended for the purpose of comparison and visibility that such a word appears in the introduction or key words.

Line 63: character space missing "ionsbetween"

Results:

- pH depth profiles in polyp and coenosarc in tissue with a high dinoflagellate density (HDD)
 - o Are there any tissue images that corroborate the location and size of the coelenteron specially in the coenosarc?
 - o What happens with the microsensor measurements after the bottom of the coelenteron? in Fig.5 there are values for the mesoglea, aboral ectoderm and ECM.
 - o Full statistical tables should appear as supplementary information.
-
- Mapping of pH in coenosarc of tissue with different alga density;
 - o Line 158 and onwards, paragraph is difficult to understand. Please rephrase.
 - o It is not clear if the measurements were performed in the same coral microcolony.
 - o The number of microcolonies used should appear when the results are presented.
 - o Line 161: "delta pH of 0.27" With respect to the value of the surface or the bottom? Please add the pH value of external seawater.
 - o Line 168: It is not clear if all of the 7 and 5 samples were subjected to both irradiance levels.
 - o Line 168: is alternatively the correct word?

Discussion:

While a recapitulation of the aims can be informative, the true discussion of the results presented by the authors only starts on the fourth paragraph, which is distracting as it reads as an extension of the introduction. The authors should address this.

A short paragraph on the implications of the results to our understanding of the impacts of ocean acidification would enrich the discussion.

Line 205: references listed talked about pH of the gastric cavity, it would be good to mention pH_{coel} refers to the same area.

Line 230: if referring to the pH of the coelenteron please use the same abbreviation as used in the rest of the manuscript (pH_{coel}).

Line 274: Reference 17 should be placed immediately after pH_{ecm} as it is unclear it is not part of the manuscript's results.

Line 288: The pH of the mesoglea is assumed to be similar to the coenosarc, what are the grounds for this assumption?

Methods

Line 336: Please provide details regarding the light quality, what type of light bulb and the instrument used to measure irradiance.

Line 339: Please provide details regarding the measuring instrument for the pH of the water bath and light.

Line 373: Please provide a rationale behind the range in the steps. Was the data pooled for the 50 micron steps profile?

Line 377: Please provide a rationale behind the range in the steps especially since the results presented in Fig.2 show steps of 25 microns. How was the data treated?

Line 380: what about under dark conditions?

References:

Species names should be in italics, please modify ref 23, 32, 44, 48, 4951, 57 and others.

Only two DOIs are listed, please revise the reference format of the journal.

Ref 10 and 24: CO₂ and O₂. The "2" should be in subscript

Figures and figure legends:

Fig.2 If the color in panel (a) would match the color for the polyp results on panel (c) (light blue) that would help identify at first hand the location difference between (a) and (b).

Fig.2 It would be instructive to mark (shadow) the coelenteron area in the panels (a) and (b)

Fig. 4 Line 600 why is symbiotic algae only represented in the oral endoderm? Is it a purely aesthetic reason or is this known? Please add reference if so.

Fig. 4 What are the values with a "?" based on?

Fig. 4 What about the aboral ectoderm pH values for LDD? where do they come from?

Reviewer #3 (Remarks to the Author):

The manuscript is well written and the science is sound. Most members of this team have been working tirelessly on coral tissue pH and this is another piece of interesting work which ties well with the rest. I recommend publication after some minor revisions as listed below.

Major points:

- I wonder if the authors considered the fact that the measurements they took at the growing edge might have been influenced by the fact that the tissue is "under construction" therefore might not have the same architecture as the rest. They did point out that polyps in that area are not fully formed which indicates the tissue is not fully formed.
- It would have been nice to have the symbiont density characterized to put things a little in perspective.
- I appreciated the mention of intracellular vesicles where the ACC is supposed to nucleate but I would have liked to see more about how this fits with their "big picture" of coral tissue pH.

Minor points: L23 "ocean's surface area" is not the right term in my view, I recommend using seabed; L42 "Symbiodiniaceae" should be in italics; L63 add a space between "ions" and "between"; It would be nice to have a figure to visualize the probe insertion and pH measurement method in general (I'm a visual person), especially for reproducibility purposes since using micro-sensors is a delicate technique.

Response to reviewers

All modifications made through the manuscript are highlighted in yellow.

Reviewer #1:

We thank Reviewer 1 for their useful comments which have helped us improve our manuscript.

General comments by reviewer #1:

- 1) *Terminology: Ectoderm and endoderm are developmental terms. Epiderm and endoderm and variants (e.g. calicoderm, gastroderm, epithelia) are the terms the authors need to substitute throughout the paper. I pointed out many but not all incidences of misuse.*

We have made the substitutions as the reviewer requests (modifications are visible throughout the manuscript). As mentioned by the reviewer, ectoderm and endoderm are usually referred to as developmental terms, but can also be used for coral adult epithelia, as used in the review by Tambutté *et al.* 2011. However, we accept the terminology used by the reviewer.

- 2) *Coelenteron = gastrovascular cavity (GVC) preferred term, is located only within the body of the polyp. The tubules/channels/extensions of the GVC that connect GVCs of adjacent polyps are not technically part of the coelenteron/GVC.*

We respectfully disagree with the reviewer on this point because we use the term coelenteron according to relevant reviews and research articles in literature. In Allemand *et al.* 2004, 2011 and Tambutté *et al.* 2011, the term coelenteron includes the gastrovascular cavity (in the polyp) and the gastrovascular canals (in the coenosarc). Another microsensor study, Yuan *et al.* 2018, and a model of coral physiology, Jokiel *et al.* 2011, also use the term coelenteron with this meaning. Regardless of which terminology is used, this cavity is a continuous space located in both the polyp and the coenosarc, as canals in the coenosarc connect the different polyps. To make sure this is clear for the reader, we have added an image of the coelenteron in the polyps and coenosarc to the manuscript as a new figure (**Figure 2b**). In addition, we have shown in our current study that the coelenteron pH is similar in polyps and coenosarc (**Figure 3c**).

- 3) *When thinking about their results the authors need to consider that corals are currently grouped into two major clades: Robusta and Complexa. Coral colonies in the Robusta clade (in which group Stylophora is placed), have well defined polyps and calices with thicker cell walls (theca) and the polyps are only connected to each other by the thin layer of coenosarc that sheets out at the tops/oral ends of the polyps. A decalcified piece of such coral looks like a sheet of naked polyps protruding downwards from the surface coenosarc that connects them (like a six pack of soda cans suspended by those plastic rings). Most of the skeletal structure in the Robusta is secreted by the calicoderm/calicoblastic epithelium located within the polyps. Thus, the GVC and its fluids are more isolated from polyp to polyp, the fluid chemistry will be more affected by the metabolic activity of that particular polyp, and this chemistry is very important to skeletogenesis in these corals. The thin coenosarc between the polyps produces the coenenchyme, which is much less dense skeletal material than the polyp calices. But the coenosarc has to keep secreting new layers of dissepiments so that the skeleton between the polyps can keep up with skeletal growth by the polyps. Corals in the Complexa clade have poorly defined corallite walls, and extensions from the GVC that connect adjacent polyps occur top to*

bottom of the polyp. A decalcified piece of a Complexa colony looks more like a kitchen sponge where the polyp mouths slightly protrude from the surface of the spongy layer, and the aboral ends of the polyps are embedded in the spongy material. Thus, there is much less isolation of the GVCs of adjacent polyps and while I have not seen measurements published, I assume there is a lot of fluid interconnection among polyps.

We thank the reviewer for their constructive comment on the distinction between robust and complex corals. *S. pistillata* belongs to the robust clade and presents heavily calcified skeleton due to the solid construction of corallite walls. Even though the polyps seem to be more isolated from each other than in a complex coral (as mentioned by the reviewer), we can clearly see from the new image that we have added to the manuscript that the coelenteron is located in both the polyps and the coenosarc. Moreover, the similarity of the coelenteron pH between polyps and coenosarc observed in the current study indicates that the fluid in these regions is homogenous despite differences in the tissue thickness of these areas. In response to the reviewer's comment, we have added sentences to the discussion that a comparative study between robust and complex corals could be of great interest for a better understanding of the regulation of coelenteric fluid in different coral species (lines 212 to 219):

*"These differences in pH depth profiles are probably linked to the tissue/skeletal relationship in these various species with some belonging to the "Complex" and other to the "Robust" clade. The latter, in which *S. pistillata* belongs, presents heavily calcified skeletons whereas the "complex" corals (e.g. *Acropora sp.*) tend to be less heavily calcified³⁷. This could have an influence on the fluid chemistry of the gastrovascular cavity with polyps being more or less isolated from each other. In this study we only focus on *S. pistillata* from the robust clade, but it would be interesting to make a comparative study on coral species belonging to the different clades."*

- 4) *Now let's think about the microcolony structure: when a planula settles, it first secretes a thin organic adhesive to attach to the substrate as it morphs into the primary polyp. All of this happens before any calcification starts, and in fact before the calicoderm differentiates. Then over days, mesenteries, a primary base plate and theca, and primary skeletal septa develop. The skirt of soft tissue around the primary polyp expands over time if the polyp has enough nourishment, and eventually more polyp mouths develop on the upper surface of this skirt. Thus, first there is tissue attachment, then tissue remodeling, then calcification/skeletogenesis. Basically, the same happens when the microcolony preps are set up. The nubbin develops a tissue skirt (coenosarc without polyp mouths and no calcification), then as the tissues differentiate you see tiny mouths pop out but without any aboral depth and little to no calcification, and finally the new skirt grows and differentiates, calicoderm forms and calcification begins. Apply this to where the present researchers took pH measurements, the HDD ones were from within polyps or coenosarc from the original nubbin (mature tissue differentiation), while the LDD ones were from the newly forming skirt without much biomass, no polyps or GVC, and most likely little to no calicoderm differentiation (would need histology to confirm).*

I make this point about the body plan and development of the species used in this study because if the authors rethink the message they are trying to present, I think their data will make for a more important and novel story. Just being the first to measure pH in the HDD vs LDD of the coenosarc of this species, whether in the light or the dark, doesn't really help us understand

anything new. This was the question I kept asking myself as I read the paper: so what?

We thank the reviewer for raising this point because it demonstrates the need for additional figure (**Figure 2**) which clarifies the histology and characteristics of HDD and LDD tissues at the growing edge. Indeed, the reviewer states that histology is needed to confirm their own interpretation. We have added this new figure to the manuscript as **Figure 2**. It does not change the interpretation of our results, but it does make the understanding of our results much clearer. In fact, calcification occurs in both HDD and LDD (growing edge) tissues where we made our measurements. Previously, the presence of growing skeleton at the growing edge is clearly shown by Venn *et al.* 2011. Additionally, at the growing edge where we made our measurements, the presence of the coelenteron and the different tissue layers are clearly visible, including the calciblastic epithelium (Venn *et al.* 2011). Non-differentiated cells are limited to the extreme distal edge (as we show in our new **Figure 2**). Measurements are not made in this zone of non-differentiated cells. We have also added a sentence to the methods (line 433-434) to make it clear to the reader that our measurements were made in an area where the skeleton is forming and the oral and aboral epithelia (including the calciderm) are present. The sentence also serves to direct the reader to the reference Venn *et al.* 2011: "In this area, the oral and aboral epithelia (including calciblastic epithelium) are present (Figure 2) and new skeleton is in the process of forming (Venn *et al.* 2011)". Together, the addition of Figure 2 and this text to the manuscript clarifies that histology of the areas that we measured.

Specific comments by reviewer #1 about the results:

- 1) *As previously demonstrated in a number of studies, GVC pH increases in the light (presumably due to photosynthesis by algal symbionts) and decreases in the dark as CO₂ and other metabolites accumulate in the GVC. Nothing new here but adding the P-I curves is very nice.*

We agree with the reviewer and we state in our discussion (line 247-248) that our results on light/dark variation of coelenteron pH are consistent with previous studies conducted on other coral species. However, the current study is the first to date to produce pH-irradiance curve to determine that a maximum coelenteron pH has been reached.

- 2) *Measuring the same light-dark pattern in the HDD coenosarc surrounding mature polyps demonstrates that the same tissue chemistry is happening in the areas in between polyps. Not a surprising result, and in fact Justin Ries used microelectrodes to measure pH in *Astrangia poculata* in 2008 and only measured between the polyps. But still nice to know that the effect of the zooxanthellae on internal pH of coral tissues. Further, much more coral animal biomass and more zoox inside the polyp so pH reached slightly higher level and closer to the mouth opening than in coenosarc. See my comment in Figure 3 below.*

As mentioned in the manuscript (**Figure 4**), coelenteron pH of both polyps and coenosarc plateaued as light intensities increased. However, at lower light intensities, coelenteron pH of the polyp is higher than in the coenosarc. This difference could be due to a higher symbiont density within the polyp, so that the pH reaches a slightly higher level than in the coenosarc. In the study of Ries *et al.* 2011 (to which we think the reviewer is referring), microsensors were also used to measure pH, but in fact these were measurements of pH in the calcifying fluid and not in the coelenteron. Our results are therefore distinct from this previous research.

- 3) *What I found more interesting, and novel was that in newly developing and poorly differentiated coral tissues, such as where they made their LDD measurements, basically there was no effect of light (because few zooxanthellae) but also most likely no calcification. Not much coral biomass per area in this region of the microcolony. But without photosynthesis, tissue/fluid pH was negative all the time. It would have been interesting if they had done a transect from the growing edge back towards the nubbin center and correlated it with the tissues differentiating to acquire zooxanthellae and the development of calicoderm and calcification.*

We fully agree that accurate quantification of symbiont density by a transect from the growing edge back to the centre of the microcolony would be better than a discrimination based only on tissue coloration. However, non-invasive methods for quantifying symbiont density in living samples are not yet available. In our current study, the difference in tissue coloration between HDD and LDD is clearly attributable to symbiont density. This allows us to make pH comparisons between the two 'extremes' of HDD and LDD. We are in the process of developing a new method to characterise symbiont density, which will help us to make the transects of gradual changes in symbiont density suggested by the reviewer in future research.

Other comments by reviewer #1:

- 1) *Again: Many issues that the authors need to address as indicated by comments and highlights in the PDF text, which I will not list here.*

We have taken into account all the comments and highlights indicated in the pdf by the Reviewer and made modifications throughout the manuscript highlighted in yellow.

- 2) *Regarding data presentation and manuscript preparation: The English needs work. I hope that one of the senior co-authors who has a better command of English can help fix many unclear and clumsy sentences and paragraphs. The discussion goes on and on, unfocused, and repeats material that should have been in methods or results.*

The manuscript has been proofread and checked by a senior co-author who has a better command of English. We have made the discussion more succinct and easier to read by removing paragraphs that repeated what was already written in the introduction, methods or results (previously lines 190 to 209): "Corals are composed of cells organized in tissues that face different compartments depending on their location [...]. In the present study, we chose to work with *Stylophora pistillata* since it is the coral species in which ECM chemistry has been most extensively studied using geochemical proxies, microsensors, or pH sensitive dyes^{6,10,13,14,17,28,29}". Another paragraph of the discussion was removed because repeating the introduction (previously lines 268 to 273): "As mentioned earlier, most calcification models consider the direct influence of the physicochemical composition of seawater on the ECM^{7,16,18-20}, [...]. Characterizing pH in these compartments is important to understanding mechanisms of proton transport linked to calcification (Figure 5)".

Specific comments by reviewer #1 about the figures:

Fig 2: Font of axis labels is way too small. Initial SW pH not shown. Use the same depth scale for the polyp and coenosarc vertical transects. Remove panel (C) and include in Figure 4.

(Now Figure 3). The figure has been modified as suggested by the reviewer (the font of the axis labels has been enlarged and the initial pH SW has been shown). For a clear understanding of the figure and to understand why it was necessary to perform these experiments, it is better to separate the pH depth profiles of polyp and cœnosarc (not the same total depth) and keep panel C in this figure. Demonstrating that there is no difference in cœlenteron pH between polyp and cœnosarc allows us to compare HDD and LDD tissues, since in the latter access to the cœlenteron is only possible through the cœnosarc.

Fig 3: The coenosarc dark and low light pH were quite a bit lower than the polyp pH. I would note this in the results. Suggests to me that the coenosarc had lower zoox density per coral biomass than in the polyp proper, and maybe only at higher light levels was pH in these inter-polyp areas affected by fluids flowing from the polyp GVC.

(Now Figure 4). As suggested by the reviewer, we added in the text of the results (lines 151 to 154): “At lower light intensities ($< 100 \mu\text{mol photons.m}^{-2}.\text{s}^{-1}$) $\text{pH}_{\text{cœl}}$, even though not statistically significantly different, is slightly higher in the polyps than in the cœnosarc which could be due to a higher density of zooxanthellae per coral biomass.”. For future research, a precise quantification of the symbiont density will be very useful.

Fig 4: Add panel (C) from Fig 2 so that all the similar type of data are together.

See comment for Fig2 **(Now Figure 3)**.

Fig 5: Rethink this model and tissue characteristics based on what I explained above. The areas where LDD measurements were made have little to no calcification going on, and I would bet, do not have a differentiated calicoderm layer, thus no calcification, ion fluxes etc.

(Now Figure 6). The presence of the cœlenteron and the different tissue layers (including the calcicoblastic epithelium) can be clearly seen in the figure that has been added to the manuscript (**Figure 2**).

Reviewer #2:

We thank Reviewer 2 for their useful comments which have helped us improve our manuscript.

Comments on the introduction by reviewer #2:

- 1) *The “polyp coelenteron” is also called, in recent literature, the polyp “gastric cavity”. It would be recommended for the purpose of comparison and visibility that such a word appears in the introduction or key words.*

As we have stated in our response to Reviewer #1, the term cœlenteron covers the gastrovascular cavity (in the polyp) and the gastrovascular canals (in the cœnosarc). In the current study we decided to use the term cœlenteron as previously used in coral calcification reviews (Allemand *et al.* 2004, 2011 and Tambutté *et al.* 2011).

- 2) *Line 63: character space missing “ionsbetween”*

Character space added for “ionsbetween” (line 65).

Comments on the results by reviewer #2:

1) *pH depth profiles in polyp and coenosarc in tissue with a high dinoflagellate density (HDD)*

o Are there any tissue images that corroborate the location and size of the coelenteron specially in the coenosarc?

The presence of the coelenteron and the different tissue layers are now shown in the figure that has been added to the manuscript (**Figure 2**). The measurements with microsensors made in the current study were performed under a microscope that allowed visual confirmation of placement of the microsensor in the coelenteron.

o What happens with the microsensor measurements after the bottom of the coelenteron? in Fig.5 there are values for the mesoglea, aboral ectoderm and ECM.

Microsensor measurements do not allow to make measurements in the mesoglea and aboral ectoderm. However, such values together with values of pH in the ECM have been obtained previously with the use of the fluorescent dye SNARF (Venn *et al.* 2022). Moreover Sevilgen *et al.* 2019 also showed that pH values in the ECM measured with SNARF match with pH values measured with microsensors. We have indicated in the legend of **Figure 6**, that values of pH in the other tissue layers (mesoglea, calicoderm and ECM) were taken from the literature with corresponding references.

o Full statistical tables should appear as supplementary information.

We have added a full statistical table as supplementary information (**Supplementary table 2**).

2) *Mapping of pH in coenosarc of tissue with different alga density*

o Line 158 and onwards, paragraph is difficult to understand. Please rephrase.

The whole paragraph “Mapping of pH_{coel} in coenosarc of tissue with different dinoflagellate density”. has been modified to improve clarity (lines 163 to 184).

o It is not clear if the measurements were performed in the same coral microcolony.

o The number of microcolonies used should appear when the results are presented.

Measurements were performed on different microcolonies: 7 samples for HDD tissues and 5 samples for LDD tissues (lines 171-173).

o Line 161: “delta pH of 0.27” With respect to the value of the surface or the bottom? Please add the pH value of external seawater.

o Line 168: It is not clear if all of the 7 and 5 samples were subjected to both irradiance levels.

o Line 168: is alternatively the correct word?

Each sample was exposed to both light and dark during time series lasting between 40 and 60 min allowing stabilization of the microsensor readings. pH value of the external SW was added. This information is provided in lines 163 to 184.

Comments on the discussion by reviewer #2:

1) *While a recapitulation of the aims can be informative, the true discussion of the results presented by the authors only starts on the fourth paragraph, which is distracting as it reads as an extension of the introduction. The authors should address this.*

We thank the reviewer for his comment on the beginning of the discussion and we have shortened and refined it (lines 194 to 201). We have made the discussion more succinct and easier to read by removing paragraphs that repeated what was already written in the introduction, methods or results (previously lines 190 to 209): “Corals are composed of cells organized in tissues that face different compartments depending on their location [...]. In the present study, we chose to work with *Stylophora pistillata* since it is the coral species in which ECM chemistry has been most extensively studied using geochemical proxies, microsensors, or pH sensitive dyes^{6,10,13,14,17,28,29}.”. Another paragraph of the discussion was removed because repeating the introduction (previously lines 268 to 273): “As mentioned earlier, most calcification models consider the direct influence of the physicochemical composition of seawater on the ECM^{7,16,18–20}, [...]. Characterizing pH in these compartments is important to understanding mechanisms of proton transport linked to calcification (Figure 5).”.

2) *A short paragraph on the implications of the results to our understanding of the impacts of ocean acidification would enrich the discussion.*

We agree with the reviewer that it is important to include ocean acidification in our discussion. As our study doesn't include any ocean acidification treatments, we are cautious not to over speculate about our findings in this context and we would prefer to mark it out as an area for future research. We have added the following paragraph to the discussion (lines 308 to 315): “The present study showed the importance of pH in proton gradients between the different coral's compartments and its implication on the calcification process. However, measurements were performed under controlled conditions and did not look at the impact of seawater acidification on the coelenteron and its possible impacts on gradients between the surrounding seawater and the coral. A study performed on *M. cavernosa* and *D. axifuga* showed a species-specific response to a decrease in seawater pH but focused only on coelenteron pH⁴. In *S. pistillata*, previous studies focused on the impact of seawater acidification on pH of the ECM^{7,15,23} but its impact on coelenteron pH remains unknown. This is an important area for future research.”

We also added a sentence at the end of the discussion (lines 328 to 331): “However, pH is not the only important parameter for calcification, and further experiments will need to fully characterize coelenteron carbonate chemistry (e.g., carbonate and calcium concentration) together with the effect of environmental parameters such as seawater acidification.”

Line 205: references listed talked about pH of the gastric cavity, it would be good to mention pH_{coel} refers to the same area.

We made this modification (line 205).

Line 230: if referring to the pH of the coelenteron please use the same abbreviation as used in the rest of the manuscript (pH_{coel}).

Discussion has been modified and changes are highlighted in yellow throughout the manuscript.

Line 274: Reference 17 should be placed immediately after pH_{ecm} as it is unclear it is not part of the manuscript's results.

We made this modification (line 263).

Line 288: The pH of the mesoglea is assumed to be similar to the coenosarc, what are the grounds for this assumption?

In **Figure 6**, the pH value in the mesoglea at the growing edge (LDD tissue) comes from the study of Venn *et al.* 2022. Since this value is similar to the value of the coelenteron pH measured in the present

study in the LDD tissue, we extrapolate that the same holds true in the HDD tissue, which makes us indicate that pH mesoglea in the HDD tissue is similar to cœlenteron pH. We have added this information in the manuscript.

Line 277 to 280: “Contrary to mesoglea pH in LDD tissue (at the growing edge²³) mesoglea pH has not been measured previously in HDD tissue in either light or dark conditions. Since in the LDD tissue, pH_{cœl} was similar to pH mesoglea, we assume that this is the same in the HDD tissue.”

Comments on the methods by reviewer #2:

Line 336: Please provide details regarding the light quality, what type of light bulb and the instrument used to measure irradiance.

Culture aquaria where coral samples are left to grow are under an irradiance of 175 $\mu\text{mol photons}\cdot\text{m}^{-2}\cdot\text{s}^{-1}$ provided by HQI light source (BLV HQI Light Bulb Nepturion, 150 W). The instrument used to measure irradiance is a Walz US-SQS/L Submersible Spherical Micro Quantum Sensor, Heinz Walz GmbH©, Germany. We added these details in the manuscript at lines 345-347.

Line 339: Please provide details regarding the measuring instrument for the pH of the water bath and light.

We provide the following information on microsensor construction and calibration: lines 357 to 359 “The pH of the seawater was measured using a pH electrode (Mettler Toledo) previously calibrated with three commercially available pH NBS buffers (pH 4, 7, 10; Hannah Instruments Buffer Solutions).”. For light measurements, we added the details in the manuscript at light 345-347.

Line 373: Please provide a rationale behind the range in the steps. Was the data pooled for the 50 micron steps profile?

We state on lines 399-400 that depth profiles in polyps were performed in incremental steps of 100 μm downward to the maximum depth. Preliminary measurements allowed us to use 100 μm steps for depth profile. As it is shown in **Figure 3a**, steps of 100 μm are precise enough to characterize pH variation in a polyp.

Line 377: Please provide a rationale behind the range in the steps especially since the results presented in Fig.2 show steps of 25 microns. How was the data treated?

We state on lines 403-404 that depth profiles in cœnosarc were performed in incremental steps of 25 μm downward to the maximum depth. Preliminary measurements allowed us to use 25 μm steps for depth profile. As it is shown in **Figure 3b**, steps of 25 μm are precise enough to characterize pH variation in a polyp.

Line 380: what about under dark conditions?

Depth profiles were not performed under dark conditions as the microsensor insertion in the polyp (through the mouth) or the cœnosarc (through the tissue) is not possible. Moreover, as mentioned in the manuscript (lines 415 to 417), “In polyps, darkness causes the polyp to retract into the corallite calyx, causing the microsensor to exit it and alter the signal.”.

Comments on the references by reviewer #2:

Species names should be in italics, please modify ref 23, 32, 44, 48, 4951, 57 and others. Only two DOIs are listed, please revise the reference format of the journal. Ref 10 and 24: CO₂ and O₂. The “2” should be in subscript.

All references were corrected as suggested by the reviewer. Concerning the DOIs, there is no obligation to add them after the references for publication in Communications Biology. To be homogeneous we have removed the only two references that mentioned a DOI.

Comments on the Figure and legends by reviewer #2:

Fig.2 If the color in panel (a) would match the color for the polyp results on panel (c) (light blue) that would help identify at first hand the location difference between (a) and (b).

Fig.2 It would be instructive to mark (shadow) the coelenteron area in the panels (a) and (b)

(Now Figure 3). We thank the reviewer for his useful comment, panel a and b were modified to make the figure easier to understand.

Fig. 4 Line 600 why is symbiotic algae only represented in the oral endoderm? Is it a purely aesthetic reason or is this know? Please add reference if so.

Fig. 4 What are the values with a “?” based on?

Fig. 4 What about the aboral ectoderm pH values for LDD? where do they come from?

(Now Figure 6). We understand that reviewer #2 was talking about **Figure 6**, i.e.the diagram depicting a model of the influence of light and dark on proton gradients across the tissue layers of a microcolony of *S. pistillata*.

We added a dinoflagellate in the aboral endoderm of LDD tissue in **Figure 6** to have the same ratio than in HDD tissue between oral and aboral endoderm. The schematic representation of dinoflagellates in **Figure 6** allows to differentiate LDD and HDD tissues.

As mentioned above for pH values in the mesoglea, values with a “?” are extrapolations of pH values measured by Venn *et al.* 2022, in the mesoglea and aboral ectoderm at the growing edge (LDD tissue). We indeed extrapolate that the calcifying cells regulate intracellular pH whatever the zone and that pH mesoglea is similar to coelenteron pH (see above). We have added this information in the manuscript.

Line 277 to 280: “Contrary to mesoglea pH in LDD tissue (at the growing edge²³) mesoglea pH has not been measured previously in HDD tissue in either light or dark condition. Since in the LDD tissue, pH_{coel} was similar to pH mesoglea, we assume that this is the same in the HDD tissue.”

Reviewer #3:

We thank Reviewer 3 for their helpful comments which have helped us improve our manuscript.

Major points by reviewer #3:

- 1) *I wonder if the authors considered the fact that the measurements they took at the growing edge might have been influenced by the fact that the tissue is "under construction" therefore might not have the same architecture as the rest. They did point out that polyps in that area are not fully formed which indicates the tissue is not fully formed.*

We understand that for Reviewer #1 and Reviewer #3 the architecture at the growing edge was not clear. We have added a new figure to the manuscript (**Figure 2**). By growing on a glass slide, the coral extends horizontally allowing measurements in both the centre of the microcolony (HDD tissues) and the growing edge (LDD tissues). In the latter, the tissue is thin but the 4 tissue layers are present within the first 400 μm from the edge of the sample where we made our measurements. In this zone, the anatomy changes with polyps starting to develop and grow. The presence of the coelenteron and the

different tissue layers can be clearly seen in the figure that has now been added to the manuscript (histological image of polyp/cœnosarc/growing edge).

- 2) *It would have been nice to have the symbiont density characterized to put things a little in perspective.*

We fully agree with the reviewer that a precise quantification of the symbiont density between the two regions measured would be better than just a discrimination based only on the tissue coloration. However, non-invasive methods for quantifying symbiont density in living samples are not yet available. The difference in tissue coloration between HDD and LDD tissues is clearly visible (from brown in the centre to transparent at the growing edge) and attributable to symbiont density. We are developing a new method for characterising the symbionts density that will help us for future research.

- 3) *I appreciated the mention of intracellular vesicles where the ACC is supposed to nucleate but I would have liked to see more about how this fits with their "big picture" of coral tissue pH.*

As for any cell, pH inside vesicles formed by macropinocytosis, should depend both on intracellular pH (that affects the physiology of the cell) and extracellular pH (since cells engulf the extracellular medium at the apical side of calcifying cells, i.e, the ECM). In our manuscript we focused on pH in the ECM and indicated (lines 296-298): "Similarly, higher pH_{ECM} would presumably also be favorable to higher pH in macropinocytotic vesicles that engulf ECM, thus potentially favoring formation of intracellular ACC (amorphous calcium carbonate) precursors in the calcicoblastic cells.". Currently we cannot discuss this further without making too many assumptions, but we hope that future work will shed some light on this aspect.

Minor points by reviewer #3:

L23 "ocean's surface area" is not the right term in my view, I recommend using seabed

We fully agree with the comment and we now accurately convey the information provided in the citation Spalding *et al.* 2001 (lines 23-24): "Coral reefs occupy less than 1.2% of the world's continental shelf area¹, but are of great ecological value."

L42 "Symbiodiniaceae" should be in italics; L63 add a space between "ions" and "between"

The corrections have been made on the manuscript and are highlighted (line 44).

It would be nice to have a figure to visualize the probe insertion and pH measurement method in general (I'm a visual person), especially for reproducibility purposes since using micro-sensors is a delicate technique.

We have added a supplementary figure (**Supplementary Figure 2**) to visualize the microsensor insertion in the polyp, the cœnosarc and at the growing edge.

REVIEWERS' COMMENTS:

Reviewer #2 (Remarks to the Author):

I believe the authors did a very good job at addressing the comments and suggestions posed in the first round of reviews. I have no further comments regarding the manuscript.

Reviewer #3 (Remarks to the Author):

I am satisfied with the edits and how the authors addressed the comments.

Reviewer #4 (Remarks to the Author):

Overall, the authors did a nice job addressing the comments of the previous reviewers and highlighting where they deviated from the opinions of the reviewers. I think the manuscript has improved and will be of great interest to coral scientists. I have several minor edits and comments that I think will further improve the big takeaways of this work to make it more clear how it fits into what is known and what is not about coral ECM pH.

Line 41: need closed parenthesis after 'polyps'

Lines 45-46: I recommend rephrasing this sentence to read less like a list of traits to improve readability.

Line 127: need comma after i.e.

Line 172: missing 'for' before 'HDD'

Lines 214-219: I would decide the way you want to reference complex and robust throughout this portion. Sometimes its in quotes and the capitalization is varied.

Line 215: need comma after e.g.

Lines 263-264: Suggest rewording this sentence for flow like so – "Our results show that pH_{coel} in LDD tissue (at the growing edge) is lower than pH_{ECM} determined in previous studies¹⁷ in light and darkness in *S. pistillata* (Figure 6)."

Line 278: Add comma after parentheses

Lines 279-280: Suggest rewording this sentence like so – "Since pH_{coel} was similar to pH_{mesoglea} in the LDD tissue, we assume that this is the same in the HDD tissue."

Line 308: Please specify what kind of pH you are referring to here like you do nicely throughout the manuscript

Lines 311-315: Please state what these studies found regarding the impact of acidification on coral pH to provide the reader some context.

Line 315: Expand on this statement to state why it is important (this will also hopefully be more clear from fleshing out the paragraph as requested above)

Lines 326-331: As the last paragraph of the manuscript, I think this could be updated to be a bit stronger. I think it would be stronger to end this manuscript with the larger implications of the work, which is still a bit difficult to take away from how it is currently written.

Methods

Line 380: I would just call this section "Scanning Electron Microscopy"

Lines 419-420: Place the 'i.e.' statements in this sentence within parentheses instead of comas here

Line 446: Please properly cite the use of R here (the software can quickly provide the correct citation information)

Figures

Figure 3: While I understand why the authors are choosing to keep the depth scales different here, I think this should be very clearly indicated in the figure caption so that readers are clearly guided here.

Figure 6: These are appearing as the same color, I suggesting making them more distinct (i.e., red and blue or something along those lines)

Response to reviewer #4

All modifications made through the manuscript are highlighted in yellow.

We thank Reviewer 4 for their useful comments which have helped us improve our manuscript.

Line 41: need closed parenthesis after 'polyps'

The corrections have been made on the manuscript and are highlighted.

Lines 45-46: I recommend rephrasing this sentence to read less like a list of traits to improve readability.

The corrections have been made on the manuscript and are highlighted: "The aboral epithelium, also known as the calicoderm, houses the calcifying cells and is located next to the skeleton, playing a key role in its formation."

Line 127: need comma after i.e.

The corrections have been made on the manuscript and are highlighted.

Line 172: missing 'for' before 'HDD'

The corrections have been made on the manuscript and are highlighted.

Lines 214-219: I would decide the way you want to reference complex and robust throughout this portion. Sometimes its in quotes and the capitalization is varied.

The corrections have been made on the manuscript and are highlighted.

Line 215: need comma after e.g.

The corrections have been made on the manuscript and are highlighted.

*Lines 263-264: Suggest rewording this sentence for flow like so – "Our results show that $pH_{c\text{œ}l}$ in LDD tissue (at the growing edge) is lower than pH_{ECM} determined in previous studies¹⁷ in light and darkness in *S. pistillata* (Figure 6)."*

The corrections have been made on the manuscript and are highlighted: "Our results show that $pH_{c\text{œ}l}$ in LDD tissue (at the growing edge) is lower than pH_{ECM} determined in previous studies¹⁷ in light and darkness in *S. pistillata* (**Figure 6**)."

Line 278: Add comma after parentheses

The corrections have been made on the manuscript and are highlighted.

Lines 279-280: Suggest rewording this sentence like so – "Since $pH_{c\text{œ}l}$ was similar to pH mesoglea in the LDD tissue, we assume that this is the same in the HDD tissue."

The corrections have been made on the manuscript and are highlighted: "Since $pH_{c\text{œ}l}$ was similar to pH mesoglea in the LDD tissue, we assume that this is the same in the HDD tissue."

Line 308: Please specify what kind of pH you are referring to here like you do nicely throughout the manuscript

The corrections have been made on the manuscript and are highlighted.

Lines 311-315: Please state what these studies found regarding the impact of acidification on coral pH to provide the reader some context.

Line 315: Expand on this statement to state why it is important (this will also hopefully be more clear from fleshing out the paragraph as requested above)

The corrections have been made on the manuscript and are highlighted (lines 311-322): “A study performed on *M. cavernosa* and *D. axifuga* showed a species-specific response to a decrease in seawater pH but focused only on coelenteron pH⁴. This study suggests that the photosynthetic activity of symbiotic dinoflagellates can partially mitigate the negative effects of ocean acidification on calcification rates. In *S. pistillata*, previous studies focused on the effects of seawater acidification on pH of the ECM^{7,15,23}. These studies have shown that ocean acidification has a major impact on coral physiology, but the effects depend on the species, light and compartment studied. The ECM is relatively well regulated with respect to pH, but mesoglea is more pH conforming with respect to the external seawater environment. However, the effects of ocean acidification on coelenteron pH remain unknown. This is an important area for future research as the coelenteron could act as a buffering compartment that mitigates the effects of decreasing pH_{SW} and helps maintain a favourable chemical environment for calcification in the ECM.”

Lines 326-331: As the last paragraph of the manuscript, I think this could be updated to be a bit stronger. I think it would be stronger to end this manuscript with the larger implications of the work, which is still a bit difficult to take away from how it is currently written.

The corrections have been made on the manuscript and are highlighted (lines 333-339): “The inclusion of coelenteron in calcification models is imperative, with particular attention to its chemical composition, especially in terms of pH. The importance lies in the efficient removal of protons from the calcification site. However, pH is only one aspect that influences calcification. For a comprehensive understanding of the coelenteron carbonate chemistry, including factors such as carbonate and calcium concentration, additional experiments are essential. Furthermore, research into the effects of environmental factors, such as seawater acidification, is crucial for a more sophisticated understanding of the calcification process.”

Methods

Line 380: I would just call this section “Scanning Electron Microscopy”

The corrections have been made on the manuscript and are highlighted.

Lines 419-420: Place the ‘i.e.’ statements in this sentence within parentheses instead of commas here

The corrections have been made on the manuscript and are highlighted.

Line 446: Please properly cite the use of R here (the software can quickly provide the correct citation information)

The corrections have been made on the manuscript and are highlighted.

Figures

Figure 3: While I understand why the authors are choosing to keep the depth scales different here, I think this should be very clearly indicated in the figure caption so that readers are clearly guided here.

The corrections have been made on the manuscript and are highlighted (now lines 614-615): “For clarity, pH depth profiles of polyp and cœnosarc are separated as the depth scales are different (not the same total depth).”

Figure 6: These are appearing as the same color, I suggesting making them more distinct (i.e., red and blue or something along those lines)

The corrections have been made on the manuscript and on the **Figure 6**.